# Occupational socialization in pre-service physical education teachers

**Dong Zhang**  *

School of Physical Education, Soochow University, Suzhou, Jiangsu, China

* zhangd0012@suda.edu.cn

## Abstract

The purpose of this study was to examine differences in occupational socialization (OS) among pre-service physical education teachers' (PPETs) in four year groups. 238 PPETs from a Chinese university. An online survey and the semi-structured interviews were used to collect data. The younger PPETs felt more prepared and believed in the benefit of professional development more than the older PPETs. The PPETs' acculturation had a critical influence on their OS. The lack of early field experience may explain why the seniors felt less confident than their younger counterparts. This study would contribute a new angle to discuss the OS and physical education teacher education program quality, especially the design of the teaching practice opportunities, together.

**Data Availability Statement:** I have updated the raw data onto Figshare. Please check them from this website: https://doi.org/10.6084/m9.figshare.20170820.v1.

**Funding:** The author received no specific funding for this work.

## Introduction

Physical education (PE) is a critical aspect of school education in which students' skills and knowledge of physical literacy, motor skills, sports, social behaviors, and self-consciousness [1] are enhanced. The physical education teacher education (PETE) program is the most important resource employed in PE teachers' training. Pre-service PE teachers (PPETs) generally spend four years completing PETE programs to qualify as PE teachers. The purpose of PETE programs is to prepare PPETs in the many aspects of school PE, including sport skills, teaching methods, and most importantly, perspectives of the PE teaching profession [2].

Lortie [3] proposed the occupational socialization (OS) theory to explain the entire process related to individuals becoming teachers. Lawson, who introduced OS to PE and PETE, defined OS as "all kinds of socialization that initially influence persons to enter the field of physical education and that later are responsible for their perceptions and actions as teacher educators and teachers" [4]. Lawson [5] posited that OS has three phases, namely, acculturation socialization, professional socialization (pre-service), and organizational socialization (in-service). Given to the study purpose, the acculturation and professional socialization phases are the most relevant. The organizational socialization phase is more focus on one's teaching career [2]. The acculturation socialization could be used to explain PETE programs' enrollment issues, to find out what caused the enrollment decline, and to recruit more students [6, 7]. As a critical factor, PPETs' background, which is a part of the acculturation socialization, impacts their decisions to choose a career in PE and orientation [8]. The students' acculturation would also affect their philosophy and understanding of the instructional model [9].

**Competing interests:** The author has declared that no competing interests exist.

Besides the backgrounds, the experiences of the PPET in the PETE programs reformed their OS about the PE profession [10]. The professional socialization phase refers to the studying period of the PETE program. In the PETE programs, the PPETs would experience the dialectical process which indicates the negotiation of the conflicts between what they learned from the K-12 PE classes and the teacher training programs [11, 12]. The PPETs carried their understanding of the PE teaching profession from their acculturation socialization phase into the PETE programs. The PE experiences from the PPET's school periods and the knowledge and skills the professors taught in the PETE program would cause the PE candidates to rethink their beliefs. Research has shown that practical teaching experience, cooperating teachers, and the PETE program faculties' philosophies had the great influence on the OS of PPETs [13, 14]. One of the most important factors to influence the PPETs' OS is the teaching practice which mostly to be carried by the methods classes and early field experience [15]. It is important to understand that the curriculum design plays a critical role to reform PPETs' orientations, either teaching or coaching. The PPETs used to have some ideas about PE before they came to the PETE programs, and many of them had a strong coaching orientation [13]. Through the teaching practices, PPETs would, usually, have a deeper understanding of the meaning of the PE profession, what the challenges a PE teacher would face, and the differences between theories and the realistic teaching situations [8].

Therefore, it is reasonable to infer that PPETs who has difference teaching practice experiences would hold the different OS developmental situations. However, limited research has been found to discuss the PPETs' OS differences. Accordingly, the purpose of this study was to examine OS differences among PPETs in four year groups. The following two research questions were formulated:

1. What are the differences in OS among PPETs in four year groups?

2. Why were their differences in OS among PPETs in four year groups?

The researcher assumed those in more advanced year groups would exhibit a mature understanding of the PE profession. This study would contribute a new angle to discuss the OS and PETE program quality, especially the design of the teaching practice opportunities, together.

## Methods

The purpose of this study was to examine OS differences among PPETs in four year groups. An online survey was employed to investigate the PPETs' OS and the results were analyzed statistically. Then, the researcher interviewed some participants to explore the deep reasons behind the survey results.

### Participants

Of the 300 PPETs in a PETE program of a university in central China, 238 completed the survey, thus yielding a response rate of 79.3%. In accordance with their year group, the participants were classified in four groups: group 1—freshman ($n = 62$), group 2—sophomore ($n = 56$), group 3—junior ($n = 54$), and group 4—senior ($n = 66$).

All participants signed the informed consent form which is a unskippable step to start the survey. At the end of the survey, the participants were asked if they were willing to be interviewed. Of the 81 participants who agreed to be interviewed, 23 participated in the group interviews: six from group 1, five from group 2, six from group 3, and six from group 4.

## Instruments

Traditionally, studies have employed qualitative methods such as interviews, observations, and documents to measure an individual's OS. Qualitative methods were deemed suitable because OS comprises personal beliefs. Adamakis and Zounhia [13] noted it is difficult to measure personal beliefs by employing quantitative methods. On the contrary, it is time-consuming and requires many human resources to employ qualitative methods in extensive investigations. Quinn [16] developed the Teachers' Occupational Beliefs Survey (TOBS) to examine the OS of pre- and in- service teachers, and established the validity by reporting the face, content, construct, and discriminant validity. The TOBS affords the opportunity to examine a large sample and conduct statistical analysis. However, the TOBS and similar questionnaires have been utilized to examine the OS of pre-service teachers, including those involved in PETE programs. The current study modified the TOBS to collect the quantitative data and used the group interviews to collect the qualitative data.

**PE-TOBS.** The quantitative data were collected by employing a modified version of TOBS. The original TOBS was developed by Quinn [16] in an endeavor to compare OS between pre- and in-service teachers. In this study, TOBS was modified to be specific for PPETs. Accordingly, the Physical Education Teachers' Occupational Beliefs Survey (PE-TOBS) was developed. The main modifications involved changing the language in the survey to accommodate PPET teachers. For example, "When you first walked into the classroom. . ." was modified to "When you first walked into the PE class. . ." and "I will be prepared to teach my subject matter" was altered to "I will be prepared to teach PE subject matter." The modifications did not affect the internal validity of PE-TOBS. Furthermore, the factor Union was removed because it was not of interest to the study.

The validity of the PE-TOBS would follow the original TOBS because there was no constructional modification was made. However, the reliability test of the TOBS was missed. To ensure the reliability of the PE-TOBS, the researcher recruited the PPETs (n = 26) from other institutions, to avoid the test effect and to complete the survey twice with a 1-week interval. Then, the researcher used the Pearson correlation coefficient to assess the correlation between the two results. All factors were significantly positively correlated at the $\alpha$ = .05 level, PCK ($r$ = .55, $p$ = .004), Control ($r$ = .41, $p$ = .003), Collaboration ($r$ = .77, $p$ = .0001), Commitment ($r$ = .78, $p$ = .0001), Preparedness ($r$ = .73, $p$ = .0001), Despair ($r$ = .58, $p$ = .002), and Development ($e$ = .54, $p$ = .004). The results indicated that the survey is reliable. Thus, the data collected from the PE-TOBS were valid and reliable.

PE-TOBS comprises six factors with 10 main questions. Multiple sub-questions constitute part of some of the main question (Table 1). Four-, five-, or six-point Likert scales were employed to evaluate the questions. The measurement principles and an example of a question of each factor are displayed in Table 1. Each factor was analyzed separately because the total score of the survey was meaningless [16]. Before the participants completed the survey, they were required to sign an informed consent form at the top of the survey.

**Interview guideline.** Semi-structured group interviews were conducted to enable the researcher to alter the interview guideline and/or ask follow-up questions [17]. The researcher designed the interview guideline based on the research purpose and previous studies. The interview aims to explore the participants' experiences to choose the PE teaching profession and their feelings of the PETE program. The guideline comprised five main questions. First, the interviewees were asked why they had majored in PE. Second, they were asked about their K-12 PE experiences. Third, the interviewees were asked about their feelings related to the PETE program. Fourth, they were asked to share what kind of PE teacher they wanted to be. Finally, they were asked how the PETE program could offer them more support.

**Table 1. PE-TOBS items.**

| Factors | Measurement Principle | Example Question |
|---|---|---|
| Control | A higher score means less control | For each action below, please indicate how much control you feel you will have during your first year of teaching. |
| Collaboration | A higher score means more collaboration | During your first year of teaching, how often per month would you like to meet with a more experienced teacher? |
| Commitment | A higher score means less commitment | Agree or disagree: I feel teaching is the most meaningful occupation I could have. |
| Preparedness | A higher score means less prepared | Agree or disagree: I will be prepared to use a variety of instructional methods. |
| Despair | A higher score means less despair | How much difficulty do you think you will have finding a teaching position in your preferred location? |
| Development | A higher score means less benefit from professional development training | Please indicate how much you feel you might benefit from the following types of professional development during your first year of teaching. |

## Procedure

The Institutional Review Board of researcher's university does not require the official application but directly approves a study like this one, which does not include any children or special needed person and obvious potential physical or phycological risk. The researcher fulfilled a form with the basic information of this study and submitted it to the academic office of the university. The researcher subsequently contacted the PETE program's department chair to grant permission to recruit their PPETs to complete the survey. Accordingly, the online survey link was sent to the 300 PPETs. As noted previously, 238 PPETs completed the survey in five days, thus yielding a response rate of 79.3%. Thereafter, group interviews were conducted with 23 of the participants. The researcher created a safe environment for the interviewees so as to protect their privacy. The interviewees were acknowledged that none of their personal information, including names, gender, majors, and classes, would be recorded. The interview recording would only retain the participants' grades and opinions as groups instead of the individuals. The four group interviews were processed in three days at different time in the first author's office. The interviewees were required not to discuss anything about the interviews to others before all interviews were completed to avoid inter-group impact. All the interviews, which were audio recorded, were transcribed so as to be analyzed.

## Data analysis

The first step involved the data screen. No missing data were reported. The researcher used the Z Residuals method by employing the range ±3.3 [18] principle to test the outlier of each factor in each group. No extreme outliers existed in any group. Multivariate Analysis of Variance (MANOVA) was utilized to test if any significant mean differences existed among the groups in relation to the six PE-TOBS factors. The alpha level was.05.

The interview data were analyzed inductively. Because the values and weight of all the data were equal, horizontalization was employed [17]. The researcher aimed to discover the narrative explanations of the survey results. The recorded interview audios were transcript into text, and the transcripts were read before employing open coding to extract all potentially useful information. Subsequently, similar codes were classified in categories. Finally, themes were extracted from categories that were connected and related. The interview data were analyzed in Chinese which is the first language of the participants. Only the themes and quoted data in this paper were translated into English by the researcher. An English-major Chinese scholar double-checked the translation to avoid potential mistakes.

## Trustworthiness

The researcher employed the PPETs from other institutions for the PE-TOBS reliability test to ensure that the PPETs in the target PETE program would not be affected by the testing effect. The researcher also processed the data screen before the data analysis. No outlier was found under $SD = \pm 3.29$. The group interviews improved the representativeness. Moreover, the researcher double-checked with the interviewees to check that there were no mistakes in the transcribed interviews. An experienced scholar was asked to review the qualitative data analysis results to ensure the reliability.

## Results

In essence, there was a significant mean difference in the four groups' PE-TOBS results. Furthermore, the participants' teaching practice experiences and acculturation socialization played critical roles in their OS development.

## Quantitative results

The researcher wished to determine if there was any significant mean difference in the PE-TOBS factors among the four groups. While the independent variable was the four groups, the dependent variables were the six PE-TOBS factors. Thus, a one-way MANOVA was appropriate.

The descriptive statistics of the PE-TOBS factors of each group are presented in Table 2. The results of the MANOVA analysis revealed there was at least one significant mean difference among the four groups. Wilk's Lambda = .862, F (18, 648.1) = 1.936, p = .011, $\eta_p^2$ = .048. Specifically, there was a significant mean difference between two PE-TOBS factors, Preparedness and Development showed significantly mean difference. Preparedness, F (3, 234) = 4.56, p = .004, $\eta_p^2$ = .055; Development, F (3, 234) = 3.907, p = .009, $\eta_p^2$ = .048.

The post-hoc test results of the two significantly different factors are displayed in Table 3. The mean of group 3 was significantly higher than those of group 1 and 2 for P*reparedness*, thus indicating that the junior PPETs felt less prepared for their future job than the freshmen and sophomores. Although the between-subject effect test results in Table 3 showed a significant mean difference for *Development*, the post-hoc test results did not show any significant difference at the .05 α level of the group comparison. The reason may be related to the fact that post-hoc tests usually over-adjust the significant level to reduce the risk of Type I errors. Thus, one may assume that p values close to .05 may indicate a significant mean difference (Warner, 2013). In this study, the researcher chose a p value of less than .07 to assume the significant

**Table 2. Descriptive statistics for PE-TOBS factors.**

| Dependent Variables | Group 1 (N = 62) | | Group 2 (N = 56) | | Group 3 (N = 54) | | Group 4 (N = 66) | |
|---|---|---|---|---|---|---|---|---|
| | *M* | *SD* | *M* | *SD* | *M* | *SD* | *M* | *SD* |
| CON* | 5.29 | 1.47 | 5.25 | 1.36 | 5.89 | 1.61 | 5.44 | 2.48 |
| COL* | 17.56 | 4.45 | 19.25 | 4.12 | 17.61 | 4.7 | 18.21 | 5.33 |
| COM* | 10.52 | 3.02 | 10.14 | 3.25 | 10.8 | 3.48 | 10.03 | 4.61 |
| PRE* | 19.98 | 6.55 | 18.89 | 5.54 | 23.48 | 6.19 | 21.06 | 8.36 |
| DES* | 29.56 | 7.24 | 30.32 | 6.33 | 29.33 | 5.63 | 27.64 | 7.68 |
| DEV* | 7.66 | 2.51 | 7.55 | 2.98 | 9.06 | 3.44 | 8.95 | 3.51 |

*. COL is Collaboration; COM is Commitment; CON is Control; DES is Despair; DEV is Development; PRE is Preparedness.

**Table 3. The multiple comparisons of preparedness and development.**

| Factor | | Mean Difference | | | |
|---|---|---|---|---|---|
| Preparedness | Class year | 1 | 2 | 3 | 4 |
| | 1 | \ | 1.09 | −3.50* | −1.08 |
| | 2 | −1.09 | \ | −4.59* | −2.17 |
| | 3 | 3.50* | 4.59* | \ | 2.42 |
| | 4 | 1.08 | 2.17 | −2.42 | \ |
| Development | Class year | 1 | 2 | 3 | 4 |
| | 1 | \ | .11 | −1.39 | −1.29 |
| | 2 | −.11 | \ | −1.50# | −1.4# |
| | 3 | 1.39 | 1.50# | \ | .10 |
| | 4 | 1.29 | 1.40# | −.10 | \ |

*. The mean difference is significant at the .05 level.

#. Assumed to be significant at the .05 level.

mean difference. Finally, the mean of group 2 was significantly lower than the means of group 3 and 4 for D*evelopment*, thus revealing the sophomore PPETs felt professional development was more beneficial than their junior and senior counterparts.

Although the researcher assumed that the PPETs in the more advanced groups would feel better prepared and believe in professional development benefits more than those in the less advanced groups, the survey data did not support this assumption. The interviews shed light on the reasons thereof.

## Qualitative results

The interview data supported the survey results. Various factors had a critical effect on the OS of the PPETs in that it affected the PPETs' acculturation, the PETE program's curriculum, and internships.

**PPETs' acculturation laid the foundation.** The level of the group was not a decisive factor in the PPETs' understanding of their profession. Rather, their background, i.e., the acculturation socialization phase, played a critical role.

When answering the question why they had majored in PE, one freshman shared, "My father is a high school PE teacher and coaches the school basketball team. I watched his work for years and knew I wanted to be like him since 15." This freshman had a clear image of what a PE teacher likes. Furthermore, he appeared to have a more enhanced notion of career choice than older PPETs. One senior admitted, "I am not sure. I just like playing soccer. They (my parents) said [with] this major (PETE), it would be easier to find a job than others."

The contrast between the freshmen and sophomore groups and junior and senior groups was most evident when they spoke about the type of PE teachers they wanted to be. While many freshmen and sophomores had goals and models to follow, some of the juniors and seniors tended to use what may be referred to as uncertain language. One freshman related, "I want to be a fun (PE) teacher, just like my middle school PE teacher." A sophomore explained, "I always felt happy in her classes. She always let us play games, even for the basketball classes. I learned lots of skills from her games. We were tired but very happy. I hope my students will love me like I love her." This PPET's middle school PE teacher was his role model and encouraged him to follow a similar path. On the contrary, a junior answered vaguely, "I do not know what kind. To be honestly, I do not know what a PE teacher really looks like. I think most of them would just give a ball and let the students play. I do not even know if I want to be a PE teacher eventually. I am here because I was not very good at the academic job."

The qualitative data concurred with the quantitative data in that the level of the group did not have the greatest impact on the OS of the PPETs. The PPETs' acculturation socialization was a foundation for their basic understanding of the PE profession and determined their major and career choices.

## PETE program's curriculum rebuilt the PPETs' beliefs

Although acculturation played a significant part in the OS of the PPETs, the experiences of participating in the PETE program, especially the curriculum, influenced their OS development. They spoke a great deal about the curriculum when they related their feelings about the program and recollected their school PE experiences.

When asked about their feelings about the PETE program, most PPETs answered positively. A participant shared, "I feel good. The professors are kind and nice. I learned a lot." Most of the juniors and seniors also mentioned the pedagogy classes. One explained,

> I did not know much pedagogical knowledge and what a professional teacher is. I used to think PE teaching was like coaching, just train the students playing sports. After taking the PE Pedagogy class, I learned so many teaching methods and I finally understood that teaching is not coaching.

The younger PPETs focused primarily on skills. One stated,

> I only play basketball. I did not know a PE teacher needs to know so many other sports. I have taken seven different skill classes, which I almost never played before. The gymnastic class was the biggest challenge and dance, yes! I never thought I was going to dance someday. But now I know I may need to teach dance in the future. You know what? I do not hate it.

Although most of the PPETs had only played one sport before they enrolled in the program, PETE majors were required to acquire more skills so as to adapt to school PE. Many of the interviewees had changed their thoughts about their school PE teachers. One related,

> After studying here (the PETE program) and thinking about my middle school PE teacher, I realized he did not do a good job. Most of the time, we just played by ourselves, the teachers only taught the basic stuff at the beginning of the semester. At that time, I was enjoying it. But by taking the pedagogy and other classes, I know a PE teacher is not supposed to have a class like that. I do not want to be like that. I want my students to learn valuable skills and knowledge from my classes.

The PPETs compared what they had learned in the program to their school PE experiences and subsequently altered their perspectives of the PE profession. Most of the PPETs mentioned the classes they took in the program when they spoke about how they had changed. The program's curriculum offered many classes that enabled the PPETs to qualify as PE teachers. Similar to the survey data, the data from the interviews did not reveal obvious group differences. All the groups were characterized by rethinking.

Some participants changed their orientations from coaching to teaching, especially the ones who had strong sport background.

> I did not like teaching PE, honestly. I thought coaching is much better than teaching. I played in both middle and high school teams. I was barely showed up in the PE classes back

in time. But, after I learned all the knowledge and skills of teaching, and practiced teaching, I fall in love with teaching PE. When you coach, you only coach one sport, but you can teach many sports and skills to the children. Besides, I do not need to focus on the competition when I teaching. All students could have fun in my class, no winners or losers. I just love that feeling.

Most of the participants showed more passions on playing sports than teaching skills. Some of the them, mostly the young PPETs, thought "PE teachers are just the ones who cannot coach because they do not play good enough." However, as long as the PPETs learned and practices more about teaching, they showed more respects to the profession, "I think teaching is more difficult. Really, I do not know how my PE teachers deal with us. So many students and things to teach."

The difference is, only the older participants showed the change in the orientations, no freshmen and sophomore did. The reason maybe the PETE program's curriculum. The PPETs in the program start to take the pedagogical courses and experience the teaching practices since the second semester of the junior year. Before that time, the curriculum was focused on the skill and content knowledge classes. The limited knowledge and experiences of PE teaching caused the youngers PPETs having few opportunities and abilities to rethink their orientation choices.

### The internship challenged the PPETs' beliefs

The PPETs' internship experiences were assessed by two questions. While the first focused on their feelings about the PETE program, the second determined their perceptions of how the PETE program could offer them more support.

The younger PPETs were optimistic. They believed in their abilities and had positive expectations of their future employment prospects. A sophomore explained,

I learned much knowledge in this program and I think I will be a great PE teacher. I know everything I need. I can do both, teaching and coaching. How difficult it could it be to teach some children? My academic grades are good and I can play many sports. I heard something like professional development. That must be helpful, right? You can communicate with other teachers and learn from each other. I would love to participate in these things.

While the freshmen and juniors expressed similar thoughts, the seniors were not optimistic. The researcher assumed that the juniors would feel more prepared during the internship because of the program. However, as related by one participant, they did not feel prepared:

The program did not give me enough training before the internship. You are not supposed to sit in the classroom for three years and be thrown into the gym directly. I had no idea what to do in my first week of teaching. Everything felt different from what I had practiced with my classmates. I think the program should have more real teaching practices before senior year, which would help us to prepare better for the internship. I do not like the professional development staff. Honestly, I do not think they (the conferences) were helpful. I was just sitting there and listened to several reports. I did not know what they [other PE teachers and experts] were talking about because I was still struggling with class management all the time.

The participants used to practice teaching with each other. "How hard it could be? I teach the PE majors all the time. Teaching the school children would be much easier." However, the

internship experience brought the real world of teaching for the PPETs. They panicked because teaching was not as they had imagined it and experienced a loss of confidence because they made so many mistakes in the gym. Moreover, a senior PPET mentioned, "I wish I could have more teaching experiences before got here (the internship). It is frustrating to how disappointed the PE teachers and professors are. I cannot stop to question myself if I chose the right path."

The internship was the first time the senior PPETs had taught in a school. They experienced conflict between their theoretical knowledge and the reality. The younger PPETs were more optimistic than the seniors because they had not experienced teaching in a school. The seniors felt less prepared and did not believe in professional development as much as their younger counterparts. This situation explained the unexpected survey result of this study, the older PPETs felt less prepared than the youngers. The participants did not have enough chance to learn from their mistakes step by step but were through into the fire without a well preparation.

## Discussion

The primary purpose of this study was to examine whether the OS of PPETs in four year groups differed. The participants completed the PE-TOBS. The results revealed significant mean differences among the four groups with regard to two factors. The researcher had hypothesized that PPETs in the more advanced groups would exhibit a more mature perspective of the PE profession than their younger counterparts. On the contrary, the younger PPETs felt more prepared and believed more in the benefits of professional development than the older PPETs. No significant mean differences were found among the four groups for any of the other factors. The qualitative data, acquired from the interviews, supported and explained the survey results.

In some aspects, there was no significant difference in the OS of the PPETs in the four groups. Their OS was influenced by their background [19] and their experiences during acculturation [6]. Ferry [10] revealed that the OS of PPETs did not change after they had participated in a field practice program and concluded that acculturation played a critical role in their OS development. This concurs with this study in that while some PPETs had taken more courses than others, they still had less knowledge about PE progression than those who had a strong PE and sports background. However, PPETs' acculturation has a continuous influence on their beliefs while participating in PETE programs [9]. Some freshmen in this study exhibited clearer goals than some seniors. One may deduce that the former had a more enhanced understanding of their mission and studied more effectively than the latter.

Furthermore, their experiences in the PETE program helped the PPETs to reconsider their ideas about their former PE teachers and experiences. Schempp and Graber [12] found that when recruits joined teacher training programs, as they learned new knowledge and skills, their ability to engage in dialectical processes was dependent on the impact of education programs on their previous beliefs. This finding concurred with those in this study. The courses, especially the pedagogy courses, helped the PPETs realize that their former PE teachers had used inappropriate methods and accordingly, the PE perspectives they developed were more advanced.

The study also explored various issues related to the PETE program curriculum, mostly about teaching practice. Most PETE programs aim to enhance PPETs' understanding of the PE profession through teaching practice [10]. Early field experience (EFE) played a critical role in building PPETs' beliefs about PE teaching [15, 20, 21]. Teaching practice helps students understand the realities of teaching and role of a PE teacher. Furthermore, it prepares PPETs

for school life and transforms them from student teachers to in-service teachers [2]. The PETE program that was assessed in this study does not implement EFE. The PPETs did not have the opportunity to practice teaching in a real environment before they underwent internship. Thus, the findings that the younger PPETs felt more confident than the seniors may have been related to optimism because they had no experience of real teaching. EFE is supposed to help PPETs and not undermine their self-confidence. Research has shown that PPETs learn teaching skills from EFE [9, 14]. However, because of a lack of EFE, the seniors in this study did not have the chance to develop self-confidence. Rather, their internship was very different to EFE. The senior PPETs were expected to teach entire PE classes at least five times per week. Their professors were not there to supervise and give them feedback. Moreover, they experienced more intense pressure during internship than if they had experienced EFE. Similar to methods classes, EFE is primarily designed to facilitate PPETs to learn from their mistakes [22]. One may deduce that the curriculum may have led to the results of this study, which were contrary to what the researcher hypothesized.

Meanwhile, unlike the traditional studies, the researcher used a survey to assess PPETs' OS. Limited study was found to use or suggest to use any quantitative method for OS assessment, except TOBS. Scholars tend to believe that OS is a subjective experience which is heavily depending on the personal thoughts of the parties and hardly to be tested objectively. However, it is necessary to build some quantitative methods, like the PE-TOBS this study used, for OS assessment because the qualitative methods would cost more time and manpower which would limit the sample size. The significant mean differences were found in this study, although the results were opposed to the hypothesis, it is still evidence to value the quantitively OS assessment methods.

The authors see this research as an inspirational study. Although there were plenty studies to exam the PPETs' OS through different lens, limited research was found to investigate the OS differences. The results of this study supported the idea that the PETE curriculum design is critical to PPETs' OS development [13, 23]. By thinking backwards, the authors asked a new question. Could PPETs' OS development reflect a PETE curriculum's quality? Currently, the curriculum design is one of the most important factors of the PETE program evaluation. Traditionally, the experts and administrators evaluated a curriculum based on the content and schedule [24]. However, the PPETs were not commonly involves in the curriculum design evaluation process. It maybe unrealistic and practically to ask the PPETs evaluating the curriculum directly but their OS developments could be a value reference. The PETE programs not only expect the PPETs to have the enough knowledge and skills, but also the mature understanding of the PE teaching profession before sending the them to the schools. Thus, the authors expected to inspire more scholars to study the relationship between PPETs' OS development and PETE curriculum design.

## Conclusion

Generally, the study found the significant differences of the PPETs' OS among different class-year groups. However, the results against the authors' assumptions. The lack of the teaching practices cause the senior PPETs feeling be flustered in their internships. The gap between the classmate-teaching practice and the field school teaching is too big to jump through without appropriate training. The dramatic contrast would make the PPETs to question themselves, either their abilities or career choices. The PETE programs should involve the PPETs' OS development into the curriculum evaluation. A practical curriculum should help the PPETs to improve their teaching skills step by step. The early field experience should start early, and the

teaching practices should be designed appropriately to increase the PPETs' confidences but not to destroy them.

## Limitations and recommendations for future studies

The participants in this study were all in the same PETE program. Thus, the results cannot be generalized to other programs. It is recommended that future studies should employ a larger sample from a number of PETE programs.

Furthermore, it is recommended that future studies should employ a longitudinal research design to explore OS development to ensure individual acculturation is controlled. This would enable an exploration of the impact of a PETE program on the OS of PPETs over a period of time.

Finally, the PE-TOBS was modified by the researcher based on an original work which is not designed for PE teachers. Future studies could build a PE focused OS survey to better reflect the pre- and in- service PE teachers.

## Acknowledgments

The researcher appreciates Dr. Ted France's help to review the interview data analysis process and Dr. Chen Liang for the language translation.

## Author Contributions

**Conceptualization:** Dong Zhang.

**Data curation:** Dong Zhang.

**Formal analysis:** Dong Zhang.

**Investigation:** Dong Zhang.

**Methodology:** Dong Zhang.

**Project administration:** Dong Zhang.

**Resources:** Dong Zhang.

**Software:** Dong Zhang.

**Writing – original draft:** Dong Zhang.

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
