## [Decision Letter · Decision Letter 0]

21 Jun 2022

PONE-D-22-11865Occupational Socialization in Pre-service Physical Education TeachersPLOS ONE

Dear Dr. Zhang,

Thank you for submitting your manuscript to PLOS ONE. After careful consideration, we feel that it has merit but does not fully meet PLOS ONE’s publication criteria as it currently stands. Therefore, we invite you to submit a revised version of the manuscript that addresses the points raised during the review process.

We look forward to receiving your revised manuscript.

Kind regards,

Ferman Konukman, Ph. D.

Academic Editor

PLOS ONE

Journal Requirements:

Reviewers' comments:

Reviewer's Responses to Questions

**Comments to the Author**

1. Is the manuscript technically sound, and do the data support the conclusions?

Reviewer #1: Yes

Reviewer #2: Yes

2. Has the statistical analysis been performed appropriately and rigorously? 

Reviewer #1: I Don't Know

Reviewer #2: Yes

3. Have the authors made all data underlying the findings in their manuscript fully available?

Reviewer #1: Yes

Reviewer #2: Yes

4. Is the manuscript presented in an intelligible fashion and written in standard English?

Reviewer #1: Yes

Reviewer #2: Yes

5. Review Comments to the Author

Reviewer #1: I acknowledge the authors effort to provide an insight in to the PETE.

The research question and the work done together with the results and findings are all coherent.

The qualitative data supporting the quantitative data strengthen the reliability of the data collected.

English language used is appropriate.

The study reveal interesting information to reconsider the PETE in some universities. In contrast to authors hypothesis, the findings will underline the importance of Early Field Experience for the Occupational Socialization in PETE.

However, the first sentence under the "Procedure" does not provide the exact information about the university that approved the study. Second, under the title "Trustworthiness", the first sentence includes an abbreviation (PCK) that was not identified anywhere else in the text.

Thank you for the effort!

Reviewer #2: First of all, I would like to thank the authors for their research work. I would like to make a few comments that I hope will help to improve their work:

1. In the abstract, write the PETE expressions clearly.

2. The methodology requires better explanations. It is not clear that this is really a mixed study. That an instrument developed for quantitative research has open-ended questions does not make it a mixed study. The authors should better justify this choice and explain the contribution of each study method to the overall research.

3. Add the average age of the PPETs in the participants section.

4. In the method section, write down the reliability values of the original scale and scale you modified.

5. Interview guideline: Explain on what basis you prepared the questions. Provide information about the reliability of the interview questions.

6. Please write more detailed information about how the qualitative analysis was done and its reliability.

6. PLOS authors have the option to publish the peer review history of their article (what does this mean?). If published, this will include your full peer review and any attached files.

Reviewer #1: **Yes: **Ertan Tufekcioglu

Reviewer #2: **Yes: **Assoc. Prof. Dr. Bijen Filiz

---

## [Author Response · Author response to Decision Letter 0]

29 Jun 2022

Dear editor and reviewers,

 I appropriate your approval and comments on my paper. I will respond to each comment/question following the sequence of the email I received. 

To the editor/Journal requirements:

1. I have modified my paper based on PLOS ONE’s sample.

2. I added some details about the ethic statement in the manuscript. Please see page 5, lines 9-10.

3. I have updated the raw data onto Figshare. Please check them from this website: https://doi.org/10.6084/m9.figshare.20170820.v1I have reviewed the references list to avoid any inappropriate citations. 

To Dr. Ertan Tufekcioglu/reviewer #1:

1. I have modified the sentences about the university’s approval of this study. Please see page 9, lines 4-9.

2. Under the Trustworthiness, “PCK” was a typo that I have modified. Please see page 11, line 3.

To Dr. Bijen Filiz/reviewer #2:

1. I have modified PETE to physical education teacher education in the abstract. Please see page 1, lines 16-17.

2. By considering your comment, I have removed the mix-methods part but explain the quantitative and qualitative research methods separately. Please see page 1, line 11-12; page 4, line 19-20; page 5, line 1-3; page 20, line 13.

3. I did not ask the participants to support their ages but only their grades. Age was not a critical impactor in my study. The participants in different grades would have different academic and teaching experiences which were considered as a critical factor to influence their OS development. 

4. I have added the information about the validity of the original and modified survey. Please see page 5, line 20; page 6, line 1. I also added the reliability test of the modified survey. Please see page 6, line 18-20; page 7, line 1-7. 

5. I added some details about the interview guideline. Please see page 8, lines 4-6.

6. I added some details about the qualitative data analysis process. Please see page 10, line 10-17; page 11, line 7-8.

I also made some modifications to the typos and other things I think should be changed. Please see the Revised Manuscript with Track Changes file. Please let me know if there is anything else I need to do.

Again, thank you so much for your help! 

Sincerely,

Dong

---

## [Editor Report · Decision Letter 1]

7 Jul 2022

Occupational socialization in pre-service physical education teachers

PONE-D-22-11865R1

Dear Dr. Zhang,

We’re pleased to inform you that your manuscript has been judged scientifically suitable for publication and will be formally accepted for publication once it meets all outstanding technical requirements.

Kind regards,

Ferman Konukman, Ph. D.

Academic Editor

PLOS ONE
---

## [Editor Report · Acceptance letter]

12 Jul 2022

PONE-D-22-11865R1 

Occupational socialization in pre-service physical education teachers 

Dear Dr. Zhang:

I'm pleased to inform you that your manuscript has been deemed suitable for publication in PLOS ONE. Congratulations! Your manuscript is now with our production department. 

Kind regards, 

on behalf of

Dr. Ferman Konukman 

Academic Editor

PLOS ONE